# Motion Flow Matching for Efficient Human Motion Synthesis and Editing

## Abstract

Human motion synthesis is a fundamental task in the field of computer animation. Recent methods based on diffusion models or GPT structure demonstrate commendable performance but exhibit drawbacks in terms of slow sampling speeds or the accumulation of errors. In this paper, we propose *Motion Flow Matching*, a novel generative model designed for human motion generation featuring efficient sampling and effectiveness in motion editing applications. Our method reduces the sampling complexity from 1000 steps in previous diffusion models to just 10 steps, while achieving comparable performance in text-to-motion and action-to-motion generation benchmarks. Noticeably, our approach establishes a new state-of-the-art result of Fréchet Inception Distance on the KIT-ML dataset. What is more, we tailor a straightforward motion editing paradigm named *trajectory rewriting* leveraging the ODE-style generative models and apply it to various editing scenarios including motion prediction, motion in-between prediction, motion interpolation, and upper-body editing.

## 1 Introduction

Human motion generation (Guo et al., 2022a; Zhu et al., 2023) constitutes a foundational task in computer animation with diverse applications spanning computer graphics, human-computer interaction, and robotics. In contrast to unconditional motion generation (Petrovich et al., 2021; Raab et al., 2023), recent endeavors have focused on introducing different conditions for enhanced controllability, such as action name (Tevet et al., 2023), text (Chen et al., 2023; Jiang et al., 2023), audio (Yi et al., 2023), and scene (Zhang et al., 2020) inputs. On the modeling front, contemporary human motion generation primarily relies on two dominant paradigms: auto-regressive methods operating in discrete spaces (Zhang et al., 2023a) and non-auto-regressive approaches grounded in diffusion models (Chen et al., 2023; Tevet et al., 2023; Zhang et al., 2022). The former often

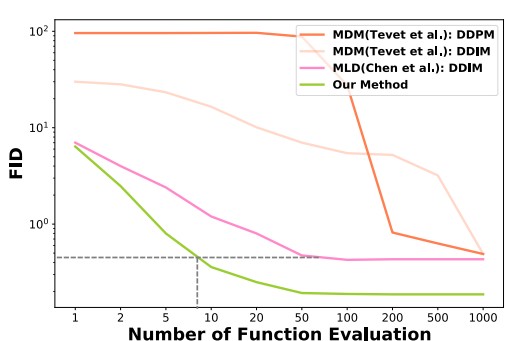

Figure 1: **FID vs. Number of Function Evaluation** on the KIT-ML dataset. Our method achieves a better FID with fewer sampling steps in comparison with baselines. Note the Y-axis is log-scaled.

accumulates errors and demands iterative frame generation, resulting in time-intensive processes. In contrast, the latter offers stability, efficient training, and seamless integration of guidance signals but is hindered by slow sampling speeds (Salimans & Ho, 2022). Quoting Chen et al. (2023): "a typical diffusion-based method MDM (Tevet et al., 2023) requires 24.74 seconds for average inference and up to a minute for maximum inference on a single V100." Although various acceleration techniques have been explored, they have yet to fundamentally alter the intrinsic curvature trajectory nature of diffusion models (Ho et al., 2020; Song et al., 2021b).

Recently, a novel generative model known as flow matching (Lipman et al., 2023; Liu et al., 2023b; Albergo & Vanden-Eijnden, 2023; Neklyudov et al., 2023) has garnered significant attention. This model is particularly effective at preserving straight trajectories during the generation process by

an ODE solver. It achieves this by regressing the linearly interpolated vector field in the training process. This makes it a promising alternative for addressing challenges related to trajectories, which are commonly encountered in diffusion models. Although flow matching has been explored in diverse domains including video (Aram Davtyan & Favaro, 2023), audio (Le et al., 2023), and point cloud (Wu et al., 2023), its application in the context of motion generation remains relatively unexplored. In this paper, we introduce the flow matching model into the task of human motion generation, remarkably, we achieve equivalent performance to previous methods that required 1000 sampling steps, but with a significantly reduced sampling time of only 10 timesteps, see Figure 1.

Moreover, recent advancements in generative models have introduced techniques for data editing and imputation, enabling the modification and restoration of data while preserving data distributions. A typical approach for image inpainting, referred to as "replacement" as highlighted in Song et al. (2021b); Ho et al. (2022), leverages the equivalence of the forward and backward passes within diffusion models to align the sampling process with known data segments and generates the unknown portions correspondingly. Nevertheless, a similar method's exploration within the context of flow matching has remained unexplored. In human motion generation, we utilize flow matching's straight trajectory property to align known motion segments with the trajectory while ensuring consistency in generating unknown motions. We provide a motion prefix and sometimes also a suffix to guide our model under textual conditions for specific motion generation that preserves consistency. Additionally, we perform inpainting in the joint space, enabling semantic editing of body parts without affecting others.

Our contributions encompass three key aspects. First, we introduce the Motion Flow Matching Model, which is a straightforward flow matching-based generative model for human motion generation. Our model strikes an optimal balance between generation quality and sampling steps across various tasks, including text-to-motion and action-to-motion generation. To the best of our knowledge, this is the inaugural application of flow matching in human motion generation. Second, we introduce a simple training-free editing method named "trajectory rewriting", facilitating editing capabilities based on flow matching, not previously explored in the existing literature. These editing techniques are versatile and well-suited for in-between motion editing, upper body manipulation, as well as motion interpolation tasks. Lastly, our experimental outcomes establish state-of-the-art Fréchet Inception Distance (FID) performance on the KIT dataset with a minimal number of sampling steps.

## 2 RELATED WORK

### 2.1 DIFFUSION AND FLOW-BASED GENERATIVE MODELS

Diffusion models (Sohl-Dickstein et al., 2015; Ho et al., 2020; Song et al., 2021b) have found broad applications in computer vision, spanning image (Rombach et al., 2022), audio (Liu et al., 2023a), video (Ho et al., 2022; Blattmann et al., 2023), and point cloud generation (Luo & Hu, 2021). Even though they have presented high fidelity in generation, they do so at the cost of sampling speed, usually demanding thousands of sampling steps. Hence, several works propose more efficient sampling techniques for diffusion models, including distillation (Salimans & Ho, 2022; Song et al., 2023), noise schedule design (Kingma et al., 2021; Nichol & Dhariwal, 2021; Preechakul et al., 2022), and training-free sampling (Song et al., 2021a; Karras et al., 2022; Lu et al., 2022; Liu et al., 2022). Nonetheless, it is important to highlight that existing methods have not fully addressed the challenge of curve trajectory modeling within diffusion models from root, as their forward pass is inherently designed to exhibit curvature in SDE, following either a linear variance schedule (Ho et al., 2020) or a cosine schedule (Nichol & Dhariwal, 2021).

A recent entrant, known as flow matching (Lipman et al., 2023; Liu et al., 2023b; Albergo & Vanden-Eijnden, 2023; Neklyudov et al., 2023), has gained prominence for its ability to maintain straight trajectories during generation by an ODE solver, positioning it as an apt alternative for addressing trajectory-related issues encountered in diffusion models. The versatility of flow matching has been showcased across various domains, including image (Lipman et al., 2023), video (Aram Davtyan & Favaro, 2023), audio (Le et al., 2023), point cloud (Wu et al., 2023), and Riemannian manifold (Chen & Lipman, 2023). This underscores its capacity to address the inherent trajectory challenges associated with diffusion models, aligning naturally with the limitations of slow sampling in the current motion generation solutions based on diffusion models.

Meanwhile, exploration of training-free editing techniques has been a significant focus within unconditional diffusion models, such as SDEdit (Meng et al., 2021) and ILVR (Choi et al., 2021), all of which rely on generative priors. Similarly, in the realm of conditional diffusion models, methods relying on cross-attention (Hertz et al., 2022; Mokady et al., 2023) have been employed. However, it's worth noting that these approaches predominantly center around SDE-based methods. Conversely, the exploration of ODE-based editing, particularly within the context of newly proposed flow matching models, remains relatively underexplored. This serves as a compelling motivation for our investigation into human motion synthesis.

## 2.2 HUMAN MOTION GENERATION

Human motion synthesis involves generating diverse and realistic human-like motion. The data can be represented using either keypoint-based (Zhang et al., 2021; Zanfir et al., 2021; Ma et al., 2023) or rotation-based (Loper et al., 2023; Pavlakos et al., 2019) representations. In this paper, we choose the rotation-based representation due to its representation efficiency resulting from the inductive bias of the human kinematic tree. In addition to unconditional motion generation, conditional inputs are used such as text (Petrovich et al., 2022; Zhang et al., 2022; Tevet et al., 2023; Guo et al., 2022b; Jiang et al., 2023), action (Petrovich et al., 2021; Guo et al., 2020; Tevet et al., 2023; Chen et al., 2023), and incomplete motion (Ma et al., 2022; Tevet et al., 2023). In this work, we mainly explore the text condition as the most informative and user-applicable medium. MDM (Tevet et al., 2023) proposes a diffusion-based generative model (Ho et al., 2020) separately trained on several motion tasks. MotionGPT (Jiang et al., 2023) presents a principled approach to the interaction between motion and language. RemoDiffuse (Zhang et al., 2023b) explores motion generation from a retrieval perspective, drawing inspiration from Blattmann et al. (2022). T2M-GPT (Zhang et al., 2023a) investigates a generative framework based on VQ-VAE and a Generative Pre-trained Transformer (GPT) for motion generation. MLD (Chen et al., 2023) advances the latent diffusion model (Rombach et al., 2022) to generate motions based on different conditional inputs. Our work introduces a novel model into human motion synthesis, with a primary objective of efficient sampling.

Additionally, the motion completion task generates motions given partial inputs, such as classical motion prediction (Zhang et al., 2021; Ma et al., 2022) or motion in-between (Tevet et al., 2023). Prior research efforts have primarily explored either SDE-style diffusion models (Tevet et al., 2023) or classical distribution alignment techniques (Ma et al., 2022). In contrast, our paper takes a novel approach by examining it from the perspective of the ODE sampling process.

## 3 METHOD

In this section, we delve into the background of the motion generation task and the novel generative models. We also introduce the framework, as illustrated in Figure 2. Finally, we present our training-free trajectory rewriting techniques, customized for the unique generative model.

### 3.1 MOTION FLOW MATCHING

Our primary goal is to synthesize a human motion $\mathbf{x}$, with a condition $\mathbf{c}$ such as a natural language description or an action label. The human motion is formed as a sequence of human poses $\mathbf{x} = \{\mathbf{x}^i\}_{i=1}^M$, where the pose in each frame $\mathbf{x}^i$ is represented by the 3D position or rotation of joints. To achieve this, we build a generative model $f$ parametrized by $\theta$ to synthesize the motion $\mathbf{x} = f(\mathbf{z}, \mathbf{c}; \theta)$, given $\mathbf{z}$ as a sampled Gaussian noise vector.

**Flow matching generation.** The generative model $f(\cdot)$ can be expressed using either an autoregressive style (Zhang et al., 2023a) or non-autoregressive models (Tevet et al., 2023). However, both approaches face challenges, such as error accumulation (Gong et al., 2022) or slow sampling speed. To alleviate those issues, we have chosen to adopt a new generative model called Flow Matching. Given a set of samples from an unknown data distribution $q(\mathbf{x})$, the goal is to learn a *flow* that pushes the simple prior density $p_0(\mathbf{x}) = \mathcal{N}(\mathbf{x} \mid 0, 1)$ towards a more complicated distribution $p_1(\mathbf{x}) \approx q(\mathbf{x})$ along the probability path $p_t(\mathbf{x})$. Formally, this is denoted using the push-forward operation as $p_t = [\phi_t]_* p_0$. Following this definition, the motion data $\mathbf{x}$ is represented as $\mathbf{x}_{t=1}$ or $\mathbf{x}_1$ while the noise vector $z$ that generates this motion is denoted as $\mathbf{x}_{t=0}$ or $\mathbf{x}_0$. The time-dependent flow can be constructed via a vector field $\mathbf{v}(\mathbf{x}, t) : \mathbb{R}^d \times [0, 1] \to \mathbb{R}^d$ that defines the flow via the neural ordinary differential equation (ODE):

$$\dot{\phi}_t(\mathbf{x}) = \mathbf{v}(\phi_t(\mathbf{x}), t), \qquad \phi_0(\mathbf{x}) = \mathbf{x}_0. \tag{1}$$

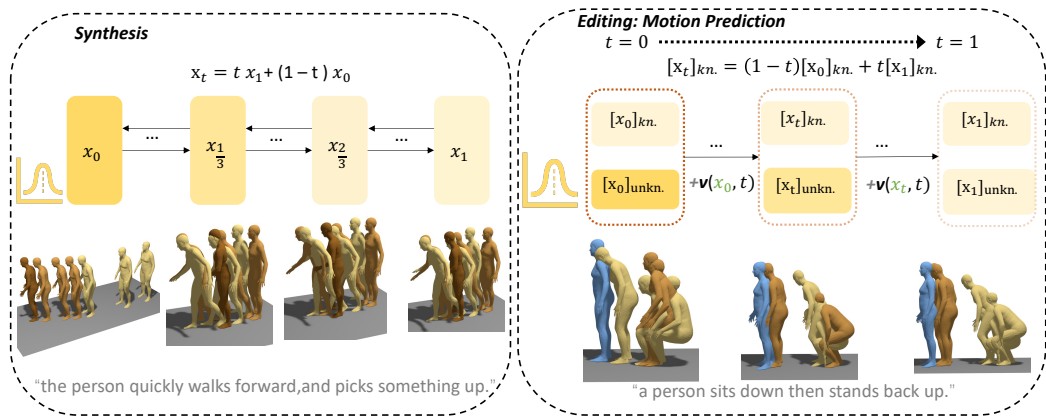

Figure 2: **Our Motion Flow Matching framework for human motion synthesis and editing.** *Synthesis*: starting with a motion feature $\mathbf{x}_1$ and a randomly sampled prior Gaussian $\mathbf{x}_0$, we gradually corrupt the motion feature using simple interpolation: $\mathbf{x}_t = t\mathbf{x}_1 + (1-t)\mathbf{x}_0$ (omitting $\sigma_{min}$ for simplicity). *Editing*: in editing tasks, we aim to synthesize unknown dimensions in the motion representation while keeping the known components. We sample a Gaussian vector $\mathbf{x}_0$ and apply trajectory rewriting to the known part of the motion as $[\mathbf{x}_t]_{\text{known}} = (1-t)[\mathbf{x}_0]_{\text{known}} + t[\mathbf{x}_1]_{\text{known}}$ while sampling an ODE solver to adapt the trajectory of the unknown part. As such, the flow $\mathbf{v}(\mathbf{x}_t, t)$ is continuously updated with the partially rewritten $\mathbf{x}_t$, enabling motion editing in various scenarios.

Given a predefined probability density path $p_t(\mathbf{x})$ and the corresponding vector field $\mathbf{w}_t(\mathbf{x})$, one can parameterize $\mathbf{v}(\mathbf{x}_t, t)$ with a neural network, parameterized by $\theta$, and solve

$$\min_{\theta} \mathbb{E}_{t, p_t(\mathbf{x})} \|\mathbf{v}(\mathbf{x}_t, t; \theta) - \mathbf{w}_t(\mathbf{x})\|^2. \tag{2}$$

**Framework.**   Noticeably, directly optimizing Equation (2) is infeasible, because we do not have access to $\mathbf{w}_t(\mathbf{x})$ in closed form. Instead, Lipman et al. (2023); Liu et al. (2023b); Albergo & Vanden-Eijnden (2023); Neklyudov et al. (2023) propose to use the conditional vector field $\mathbf{w}_t(\mathbf{x} \mid \mathbf{x}_1)$ as the target, which corresponds to the conditional flow $p_t(\mathbf{x} \mid \mathbf{x}_1)$. Importantly, they show that this new *conditional Flow Matching* objective

$$\min_{\theta} \mathbb{E}_{t, p_t(\mathbf{x} \mid \mathbf{x}_1), q(\mathbf{x}_1)} \|\mathbf{v}(\mathbf{x}_t, t; \theta) - \mathbf{w}_t(\mathbf{x} \mid \mathbf{x}_1)\|^2, \tag{3}$$

has the same gradients as Equation (2). By defining the conditional probability path as a linear interpolation between $p_0$ and $p_1$, all intermediate distributions are Gaussians of the form $p_t(\mathbf{x} \mid \mathbf{x}_1) = \mathcal{N}(\mathbf{x} \mid t\mathbf{x}_1, 1 - (1 - \sigma_{\min})t)$, where $\sigma_{\min} > 0$ is a small amount of noise around the sample $\mathbf{x}_1$: $\mathbf{x}_t = t\mathbf{x}_1 + [1 - (1 - \sigma_{\min})t]\mathbf{x}_0$. The corresponding target vector field is:

$$\mathbf{w}_t(\mathbf{x} \mid \mathbf{x}_1) = \mathbf{x}_1 - (1 - \sigma_{\min})\mathbf{x}. \tag{4}$$

Learning the straight trajectory improves the training and sampling efficiency compared to diffusion paths. When we need extra condition signals $\mathbf{c}$, we can directly insert them into the vector field estimator $\mathbf{v}(\mathbf{x}_t, t, \mathbf{c}; \theta)$. Overall, the framework of flow matching allows it to generate samples by first sampling $\mathbf{x}_0 \sim \mathcal{N}(\mathbf{x} \mid 0, 1)$ and then solving Equation (1) using an off-the-shelf numerical ODE solver (Runge, 1895; Kutta, 1901; Alexander, 1990). In the end, we can formulate $f$ as $\mathbf{x} = f(\mathbf{x}_0) = \text{ODESolve}(\mathbf{x}_0, \mathbf{c}; \theta)_{t:0 \to 1}$.

**Sampling.**   After the training of the neural velocity field $\mathbf{v}(\mathbf{x}_t, t, \mathbf{c}; \theta)$, the generation of samples is facilitated through practical discretization of the ordinary differential equation (ODE) process outlined in Equation (1) by employing an ODE solver. Using the Euler ODE solver as an illustration, this discretization method entails dividing the process into $N$ steps, resulting in the following expression:

$$\mathbf{x}_{(\hat{t}+1)/N} \leftarrow \mathbf{x}_{\hat{t}/N} + \frac{1}{N}\mathbf{v}(\mathbf{x}_{\hat{t}/N}, \frac{\hat{t}}{N}, \mathbf{c}; \theta), \tag{5}$$

where the integer time step $\hat{t} = 0, 1, \cdots, N - 1$ such that $t = \hat{t}/N$. Additionally, for enhanced efficiency in sampling, alternative approaches such as adaptive step-size ODE solvers Runge (1895); Kutta (1901) can be considered, which can significantly reduce the computational time required.

Given that vector field regression in flow matching emulates noise prediction techniques used in diffusion models (Zheng et al., 2023), we further investigate the incorporation of classifier-free guidance (Ho & Salimans, 2021) in flow matching. This entails introducing random dropout to the conditional signals. In practice, our network learns both the conditioned and unconditioned distributions by randomly setting $\mathbf{c} = \emptyset$ for 10% of the samples. This configuration effectively causes $\mathbf{v}(\mathbf{x}_t, t, \emptyset; \theta)$ to approximate $p(\mathbf{x}_1)$, signifying that a predominant portion of the network's capacity is dedicated to conditional sampling (90%) rather than unconditional sampling (10%). Subsequently, we conduct the sampling according to the equation:

$$\mathbf{v}_s(\mathbf{x}_t, t, \mathbf{c}; \theta) = \mathbf{v}(\mathbf{x}_t, t, \emptyset; \theta) + s \cdot (\mathbf{v}(\mathbf{x}_t, t, \mathbf{c}; \theta) - \mathbf{v}(\mathbf{x}_t, t, \emptyset; \theta)), \qquad (6)$$

where $s$ indicates the guidance strength that balances the trade-off between diversity and fidelity.

**The network.** The dynamics of the flow matching model $\theta$ is founded upon a simple encoder-only architecture based on the Transformer (Vaswani et al., 2017). The Transformer architecture is meticulously designed to possess temporal awareness, facilitating the acquisition of knowledge pertaining to motions of varying durations. Its efficacy within the motion domain has been empirically substantiated (Tevet et al., 2023; Duan et al., 2021; Aksan et al., 2021). For model inputs, $\mathbf{x}$, the time-step $t$, and the condition code $\mathbf{c}$, undergo individual fully connected projections into the Transformer dimension via feed-forward networks. These projections are subsequently aggregated to yield the token $\mathbf{h}_{[x_t,t,c]}$. Each frame of the noisy input data $\mathbf{x}_t$ is subject to linear projection into the Transformer dimension and is summed with a positional embedding. The detailed structure can be found in Appendix Figure 10.

### 3.2 MOTION EDITING

**Editing task.** We mainly explore the following editing operations: 1) Motion in-between based on prefix and suffix in the temporal domain, 2) Motion prediction based on prior prefix motion in the temporal domain, 3) Motion interpolation with a gap of frames, and 4) Editing body parts in the spatial domain. The editing operations involve only the sampling process in inference, without any additional training steps. In temporal editing (in-between and motion prediction), the input consists of the prefix and suffix frames of the motion sequence. In the spatial setting, we hope that body parts can be re-synthesized while preserving the rest of the body. Editing can be performed conditionally or unconditionally, with the option to set $\mathbf{c} = \emptyset$ in the latter case.

**Motion editing by trajectory rewriting.** In diffusion models, a technique known as "replacement" is employed during the sampling process to address data imputation challenges (Ho et al., 2022; Song et al., 2021b). The network gradually corrupts data following either a variance-preserving (VP-SDE) (Ho et al., 2020) or variance-explosion (VE-SDE) approach (Song et al., 2021b). In contrast, flow matching takes a fundamentally different approach such that it initially knows the target distribution sample $\mathbf{x}_0$ and then "causalizes" the intermediate data sample $\mathbf{x}_t$ through simple interpolation: $\mathbf{x}_t = t\mathbf{x}_1 + [1 - (1 - \sigma_{min})t]\mathbf{x}_0$. This stands in stark contrast to diffusion models, where $\mathbf{x}_0$ becomes known only after conducting sufficient *stochastic* forward steps to sample it.

Editing tasks aim to preserve known parts during editing while generating the unknown parts in a manner consistent with the known ones. Formally, let $\mathcal{M}$ denote a binary boolean mask with the same dimensions as the motion representation $\mathbf{x}_1$, such that $\mathcal{M} = 1$ indicates known corresponding dimension in the given motion $\mathbf{x}_1$, and $\mathcal{M} = 0$ otherwise. Utilizing the principle of pursuing a straight trajectory in flow matching, we consistently enforce the known dimensions as the linear interpolation between $\mathbf{x}_0$ and $\mathbf{x}_1$ during sampling steps, while adapting the trajectory of the unknown dimensions from noise. Specifically, in contrast to the standard process for motion synthesis in Equation (5), the sampling process for editing is formalized as:

$$\tilde{\mathbf{x}}'_{\hat{t}/N} \leftarrow \underbrace{(1 - \mathcal{M}) \cdot \tilde{\mathbf{x}}_{\hat{t}/N}}_{=[\tilde{\mathbf{x}}'_{\hat{t}/N}]_{\text{unknown}}} + \underbrace{\mathcal{M} \cdot \left( (1 - \frac{\hat{t}}{N})\mathbf{x}_0 + \frac{\hat{t}}{N}\mathbf{x}_1 \right)}_{=[\tilde{\mathbf{x}}'_{\hat{t}/N}]_{\text{known}}}, \qquad (7)$$

$$\tilde{\mathbf{x}}_{(\hat{t}+1)/N} \leftarrow \tilde{\mathbf{x}}'_{\hat{t}/N} + \frac{1}{N} \mathbf{v}(\tilde{\mathbf{x}}'_{\hat{t}/N}, \frac{\hat{t}}{N}, \mathbf{c}; \theta). \qquad (8)$$

where $\mathbf{x}_0$ is a random sampled Gaussian noise that aims to match with the target data $\mathbf{x}_1$. The intermediate results are denoted with $\tilde{\mathbf{x}}_{\hat{t}/N}$ to discriminate from standard sampling in Equation (5), and $\tilde{\mathbf{x}}'_{\hat{t}/N}$ denotes the manipulated $\tilde{\mathbf{x}}_{\hat{t}/N}$ during the sampling process.

---

**Algorithm 1** Euler Sampling algorithm with Trajectory Rewriting.

---

1: **Input**: $\mathbf{x}_1$ the original motion (or partial data with valid known dimensions); $\mathcal{M}$ the boolean mask indicating known / unknown dimensions in the motion; $\mathbf{v}$ and $\theta$ the vector field predictor with pretrained parameters
2: **Parameters**: $N$ the number of sampling steps; $\varsigma$ the threshold when trajectory rewriting stops.
3: Sample $\mathbf{x}_0 \sim \mathcal{N}(0,1)$ from the Gaussian distribution, $\tilde{\mathbf{x}}_0 = \mathbf{x}_0$ at $\hat{t} = 0$.
4: **for** $\hat{t} = 1, 2, ..., N-1$ **do**
5:     **if** $\frac{\hat{t}}{N} < \varsigma$ **then**
6:         Rewrite $\tilde{\mathbf{x}}'_{\hat{t}/N} \leftarrow (1 - \mathcal{M}) \cdot \tilde{\mathbf{x}}_{\hat{t}/N} + \mathcal{M} \cdot \left( (1 - \frac{\hat{t}}{N}) \mathbf{x}_0 + \frac{\hat{t}}{N} \mathbf{x}_1 \right)$
7:         $\tilde{\mathbf{x}}_{(\hat{t}+1)/N} \leftarrow \tilde{\mathbf{x}}'_{\hat{t}/N} + \frac{1}{N} \mathbf{v}(\tilde{\mathbf{x}}'_{\hat{t}/N}, \frac{\hat{t}}{N}, \mathbf{c}; \theta)$.
8:     **else**
9:         $\tilde{\mathbf{x}}_{(\hat{t}+1)/N} \leftarrow \tilde{\mathbf{x}}_{\hat{t}/N} + \frac{1}{N} \mathbf{v}(\tilde{\mathbf{x}}_{\hat{t}/N}, \frac{\hat{t}}{N}, \mathbf{c}; \theta)$.
10:     **end if**
11: **end for**
12: **Return**: The motion after editing $\tilde{\mathbf{x}}_{N/N} = \tilde{\mathbf{x}}_1$.

---

Table 1: **Comparison with state-of-the-art methods on the KIT-ML** (Plappert et al., 2016) test set. RP Top3 denotes R-Precision Top3. NFE denotes the number of function evaluations. $\rightarrow$ indicates that closer to real is better.

| Methods | NFE ↓ | RP Top3↑ | FID ↓ | MM-Dist ↓ | Diversity → | MModality ↑ | #params |
|---|---|---|---|---|---|---|---|
| Real motion | - | $0.779^{\pm.006}$ | $0.031^{\pm.004}$ | $2.788^{\pm.012}$ | $11.08^{\pm.097}$ | - | |
| TM2T Guo et al. (2022b) | - | $0.587^{\pm.005}$ | $3.599^{\pm.153}$ | $4.591^{\pm.026}$ | $9.473^{\pm.117}$ | $\mathbf{3.292}^{\pm.081}$ | 317M |
| Guo et al. (2022a) | - | $0.681^{\pm.007}$ | $3.022^{\pm.107}$ | $3.488^{\pm.028}$ | $10.72^{\pm.145}$ | $2.052^{\pm.107}$ | 181M |
| T2M-GPT | - | $0.716^{\pm.006}$ | $0.737^{\pm.049}$ | $3.237^{\pm.027}$ | $11.198^{\pm.086}$ | $2.309^{\pm.055}$ | 247.6M |
| MDM Tevet et al. (2023) | 1,000 | $0.396^{\pm.004}$ | $0.497^{\pm.021}$ | $9.191^{\pm.022}$ | $10.847^{\pm.109}$ | $1.907^{\pm.214}$ | 23M |
| MotionDiffuse | 1,000 | $\mathbf{0.739}^{\pm.004}$ | $1.954^{\pm.062}$ | $\mathbf{2.958}^{\pm.005}$ | $11.100^{\pm.143}$ | $0.730^{\pm.013}$ | 238M |
| MLD Chen et al. (2023) | 50 | $0.734^{\pm.007}$ | $0.404^{\pm.027}$ | $3.204^{\pm.027}$ | $10.800^{\pm.117}$ | $2.192^{\pm.071}$ | 26.9M |
| **Our MFM** | **10** | $0.414^{\pm.006}$ | $\mathbf{0.359}^{\pm.034}$ | $9.030^{\pm.043}$ | $11.310^{\pm.102}$ | $1.220^{\pm.079}$ | 17.9M |
| **Our MFM** | **50** | $0.415^{\pm.006}$ | $\mathbf{0.193}^{\pm.020}$ | $9.041^{\pm.013}$ | $\mathbf{11.080}^{\pm.108}$ | $1.490^{\pm.056}$ | 17.9M |

In our experiments, we found that editing operations do not need to be employed throughout the entire ODE sampling process. Restricting the trajectory rewriting operation only in early time steps suffices to ensure consistent generation, while granting us more flexibility for editing. Specifically, we set $\varsigma \in [0, 1]$ as a threshold such that the rewriting is only applied before that step $t = \frac{\hat{t}}{N} < \varsigma$. Empirically we set $\varsigma = 0.2$ throughout this work. The complete process trajectory rewriting for Euler sampling is shown in Algorithm 1. Under the property of a straight trajectory, the completed unknown part of the motion correctly exhibits the desired marginal distribution by design, and naturally aligns with the known part due to the fully optimized vector field estimator.

## 4 EXPERIMENT

### 4.1 DATASETS AND EXPERIMENTAL DETAILS

Our experimental evaluations are conducted on three established datasets commonly employed for human motion generation tasks: HumanML3D (Guo et al., 2022a), KIT Motion-Language (KIT-ML) (Plappert et al., 2016) for text-to-motion generation, and an additional action-to-motion generation dataset, HumanAct12 (Guo et al., 2020). We adhere to the evaluation protocols outlined in Guo et al. (2022a). We opt for a motion representation following Guo et al. (2022a) for its effectiveness in encoding the motion kinematics. More details are introduced in Appendix E.2. Similar to Guo et al. (2022a), the dataset KIT-ML and HumanML3D are extracted into motion features with dimensions 251 and 263 respectively, which correspond to local joints position, velocity, and rotations in root space as well as global translation and rotations. These features are computed from 21 and 22 joints of SMPL (Loper et al., 2023).

Our evaluation metrics encompass five key aspects. 1). We evaluate the general number of function evaluations (NFE), denoting the average network forward number. 2), To assess the parameter efficiency of the models, we investigate the number of parameters they contain. 3), For motion quality assessment, we rely on the Frechet Inception Distance (FID), utilizing a feature extractor (Guo et al., 2022a) to measure the distance between feature distributions of generated and real motions. 4), To

Table 2: **Comparison with state-of-the-art methods on the HumanML3D** (Guo et al., 2022a) test set. RP Top3 denotes R-Precision Top3. NFE denotes the number of function evaluations. → indicates that closer to real is better.

| Methods | NFE ↓ | RP Top3↑ | FID ↓ | MM-Dist ↓ | Diversity → | MModality ↑ | #params |
|---|---|---|---|---|---|---|---|
| Real motion | - | $0.797^{\pm.002}$ | $0.002^{\pm.000}$ | $2.974^{\pm.008}$ | $9.503^{\pm.065}$ | - | |
| TM2T Guo et al. (2022b) | - | $0.729^{\pm.002}$ | $1.501^{\pm.017}$ | $3.467^{\pm.011}$ | $8.589^{\pm.076}$ | $2.424^{\pm.093}$ | 317M |
| Guo et al. (2022a) | - | $0.736^{\pm.002}$ | $1.087^{\pm.021}$ | $3.347^{\pm.008}$ | $9.175^{\pm.083}$ | $2.219^{\pm.074}$ | 181M |
| T2M-GPT | - | $0.685^{\pm.003}$ | $\mathbf{0.140}^{\pm.006}$ | $3.730^{\pm.009}$ | $9.844^{\pm.095}$ | $\mathbf{3.285}^{\pm.070}$ | 247.6M |
| MotionGPT | - | $0.778^{\pm0.002}$ | $0.232^{\pm.008}$ | - | $\mathbf{9.520}^{\pm.071}$ | $2.008^{\pm.084}$ | 220M |
| MDM Tevet et al. (2023) | 1,000 | $0.611^{\pm.007}$ | $0.544^{\pm.044}$ | $5.566^{\pm.027}$ | $9.559^{\pm.086}$ | $2.799^{\pm.072}$ | 23M |
| MotionDiffuse | 1,000 | $\mathbf{0.782}^{\pm.001}$ | $0.630^{\pm.001}$ | $\mathbf{3.113}^{\pm.001}$ | $9.410^{\pm.049}$ | $1.553^{\pm.042}$ | 238M |
| MLD Chen et al. (2023) | 50 | $0.772^{\pm.002}$ | $0.473^{\pm.013}$ | $3.196^{\pm.000}$ | $9.724^{\pm.082}$ | $2.413^{\pm.079}$ | 26.9M |
| **Our MFM** | **10** | $0.642^{\pm.003}$ | $0.362^{\pm.006}$ | $5.280^{\pm.009}$ | $9.860^{\pm.095}$ | $2.443^{\pm.070}$ | 17.9M |

Table 3: **Evaluation of trajectory rewriting editing.** We edit our motion by randomly generating $5,000$ motions, and compare it with the ground truth. ADE and FDE are joint distances between generation and ground truth.

| | Prediction | | | Upper body | | In-between | |
|---|---|---|---|---|---|---|---|
| | FID ↓ | ADE↓ | FDE↓ | FID↓ | ADE↓ | FID↓ | ADE↓ |
| MDM Tevet et al. (2023) | 7.34 | 5.90 | 7.50 | 8.40 | 5.40 | 3.43 | 4.73 |
| **Our MFM** | **5.79** | **4.99** | **5.50** | **6.46** | **4.12** | **2.59** | **3.32** |

gauge generation diversity, we employ the Diversity metric, which quantifies motion diversity by calculating variance in features extracted from the motions, along with MultiModality (MModality) for assessing diversity within generated motions under the same text description. 5), In terms of text alignment, we utilize the motion-retrieval precision (R Precision) to evaluate the accuracy of matching texts and motions using Top3 retrieval accuracy, while Multi-modal Distance (MM Dist) measures the distance between motions and texts, all based on feature spaces from Guo et al. (2022a).

We use AdamW (Loshchilov & Hutter, 2019) optimizer with $[\beta_1, \beta_2] = [0.9, 0.999]$, batch size of 256. We train with a learning rate of 1e-4. we employ a timestep of $N = 10$ for Euler ODE sampling. However, for trajectory rewriting, we opt for a slightly larger value of $N = 30$, but restrict the editing to the initial $t = 0.2$ timesteps, effectively modifying the first 6 timesteps only. More details about implementation and evaluation metrics are provided in the Appendix.

## 4.2 MAIN RESULT

**Text-to-Motion.** In the text-to-motion generation, we present our results on the KIT dataset in Table 1 and on the HumanML3D dataset in Table 2. Our approach attains state-of-the-art performance in FID on the KIT dataset while requiring minimal function evaluations and a modest number of parameters. These tables clearly highlight the successful achievement of a favorable balance between sampling steps (NFE) and generation performance by our method. Notably, GPT-based methods such as T2M-GPT (Zhang et al., 2023a) and MotionGPT (Jiang et al., 2023), which rely on token prediction, tend to require a higher number of network forward evaluations (NFEs), equivalent to the number of tokens, compared to our approach, which uses 10 NFEs. This suggests that our method may be more computationally efficient in terms of NFEs required for motion generation.

In Figure 3, we offer a qualitative comparison with three baseline methods: MDM (Tevet et al., 2023), MLD (Chen et al., 2023), and T2M-GPT (Zhang et al., 2023a), where we use the same guidance strength $s = 2.5$. Our results exhibit enhanced capabilities in capturing the nuanced motion dynamics derived from the input prompts as compared to these baseline approaches. For additional qualitative results in text-to-motion synthesis, please refer to Appendix Figure 8.

**Sampling steps.** In Figure 1, We investigate the relationship between the number of sampling steps and FID (Fréchet Inception Distance) using the following baselines on the KIT-ML dataset. 1). MDM (Tevet et al., 2023) with DDPM sampling. 2). MDM with DDIM sampling (Song et al., 2021a). 3). MLD Chen et al. (2023) with DDIM sampling. MDM fails to achieve reasonable FID due to the design. Our method converges to a lower FID at the same sampling steps. Our approach is also significantly faster than MLD (Chen et al., 2023) and MDM (Tevet et al., 2023),

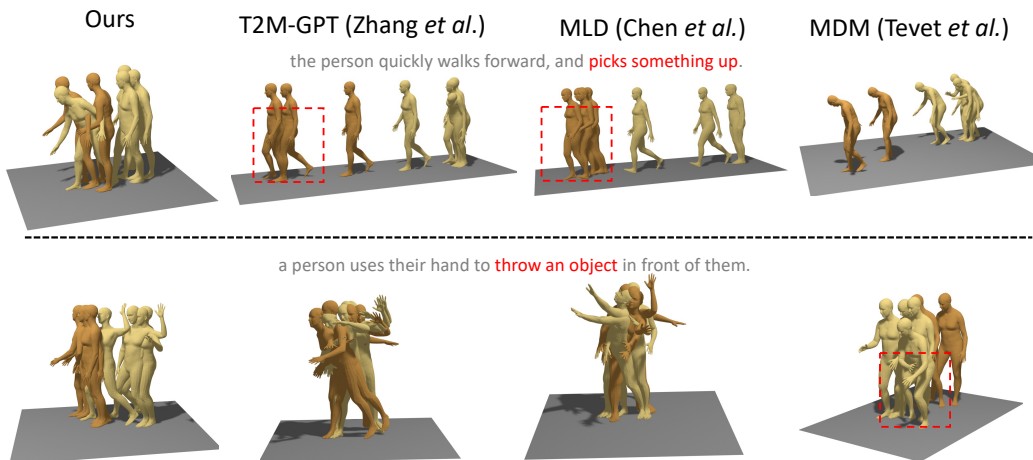

Figure 3: **Qualitative comparison with baselines.** The generated motion is represented by the bronze frames. Please take note of the dotted red rectangle; occasionally, our baseline may struggle to discern the guidance from the prompt, resulting in less fluid motion.

being approximately 5 times faster than MLD and 100 times faster than MDM. It is worth noting that our Transformer-based architecture can be further optimized using FlashAttention (Dao et al., 2022).

**Action-to-Motion.** We further showcase our action-to-motion generation results in Appendix Table 4. Our method consistently achieves on-par results with baselines, while requiring significantly fewer function evaluations and better parameter-efficiency, underscoring the efficacy of our approach.

### 4.3 MOTION EDITING BY TRAJECTORY REWRITING

**Qualitative result.** In Figure 4, we discover that editing operations can effectively take place within the previous 0.2 time steps other than the full 1.0 time steps. Furthermore, we provide a demonstration of $x_1$ estimation at intervals of 0.1 time steps. Remarkably, we observe that even the initial $x_t$ can yield reasonably accurate estimations of $x_1$. As time progresses, these estimations gradually align more closely with the provided prompt, eventually stabilizing around $t = 0.2$. This phenomenon underscores the straight trajectory characteristics of our models.

Moreover, we investigate several editing operations including in-between, future prediction, upper body, and interpolation editions, as illustrated in Figure 5. We examine the alteration of the text prompt while retaining the known part, serving as a test to evaluate whether our generative models can consistently produce motion that aligns with the preserved known segment.

**Quantitative result.** In our comparison with the baseline method MDM (Tevet et al., 2023) as illustrated in Table 3, we evaluate our trajectory rewriting approach through identical editing operations, revealing its slight performance superiority across FID, Average Displacement Error (ADE), and Final Displacement Error (FDE) metrics, commonly used in motion prediction studies (Zhang et al., 2021). More details can be found in Appendix D.

## 5 CONCLUSION

In this work, we have introduced a straightforward yet highly effective generative model called Flow Matching to the realm of human motion synthesis. Our results demonstrate a remarkable balance between generation fidelity and sampling steps. Leveraging the inherent property of straight trajectories, we have devised a simple trajectory rewriting technique for training-free editing. In our future endeavors, we intend to extend this trajectory rewriting technique to other domains, such as image editing.

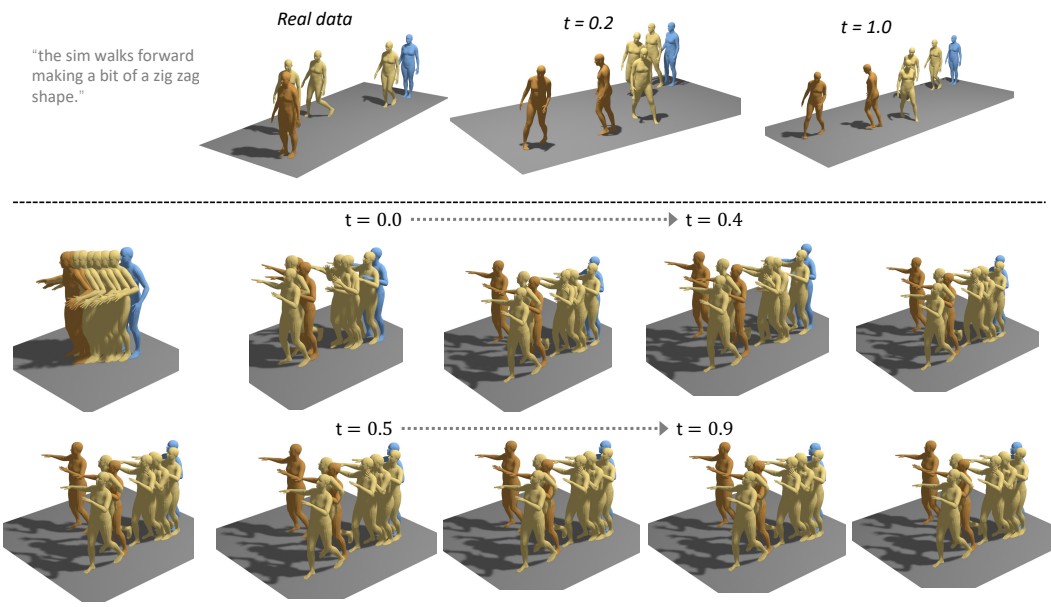

Figure 4: **Above is a comparison of editing times for the motion prediction task.** Rewriting the trajectory up to $t = 0.2$ achieves nearly identical performance compared to rewriting up to $t = 1.0$. **Below is the motion prediction: estimation of $\mathbf{x}_1$ during trajectory rewriting from $t = 0.0$ to** $t = 1.0$ It's worth noting that from the very first time steps, the model can already produce reasonably accurate motion. Furthermore, as time progresses, the estimation of $\mathbf{x}_1$ gradually aligns more closely with the provided prompt. Remarkably, by the time we reach time step $t = 0.2$, the generated human motion exhibits a high level of alignment with the prompt. Light blue frames denote motion input, while bronze frames signify generated motion. The gradient of colors, ranging from light to dark, signifies the passage of time.

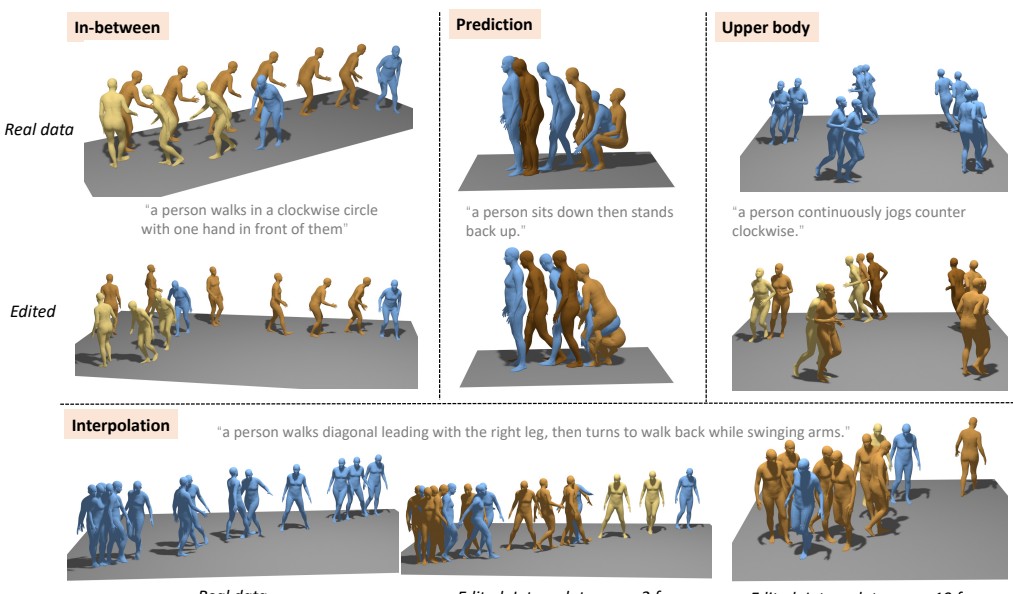

Figure 5: **Motion Editing by trajectory rewriting.** We focus on showcasing four key editing scenarios in human motion generation: 1) In-between editing, 2) Motion prediction from a partial sequence, 3) Upper body editing while keeping lower body joints fixed, and 4) Interpolating missing motion frames using specified motions. Additionally, we conduct rotations of the views to obtain a more comprehensive global perspective.

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

# A APPENDIX

## CONTENTS

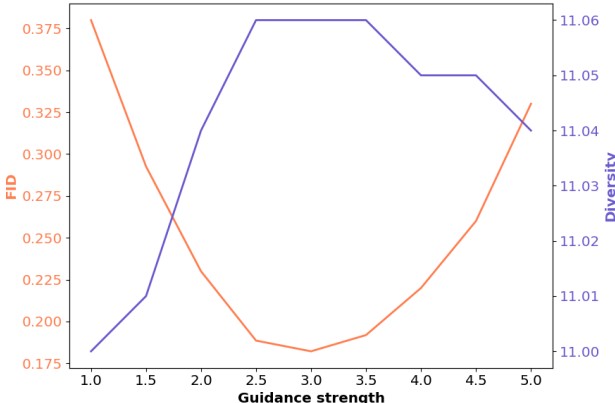

Figure 6: **The FID and Diversity on the guidance strength** of our method on KIT-ML dataset.

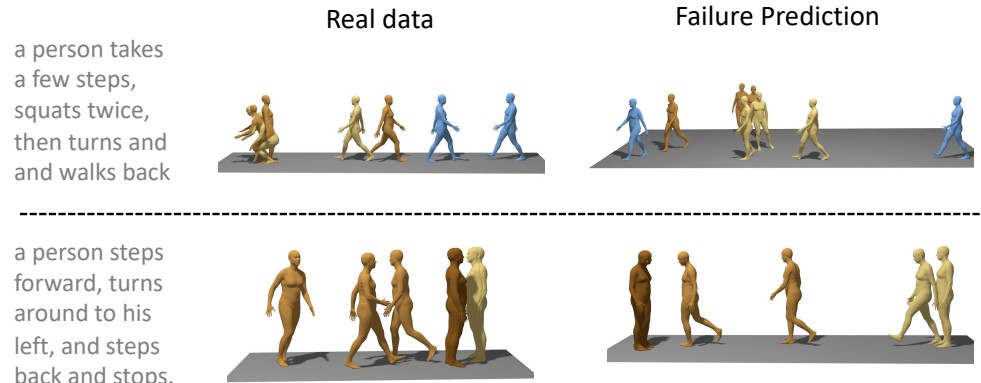

Figure 7: **Failure case visualization** of motion in-between editing and text-to-motion synthesis on HumanML3D dataset.

## B SOCIETAL IMPACT

This work provides a powerful tool for quickly generating highly realistic human pose series. It has the potential to significantly expedite the process of creating human pose-related artwork, thus enabling the democratization of creativity. However, on the flip side, this tool also poses a challenge to the community in terms of creating safety mechanisms to prevent the misuse of generative models for deep-fake and misleading content creation with potentially controversial intentions.

## C MORE RESULTS

### C.1 QUALITATIVE RESULTS

**Guidance Strength.** We also explore the guidance strength, denoted as $s$, in the classifier-free guidance, as depicted in Figure 6 on dataset KIT-ML. Interestingly, we observe that a value of $s = 3.0$ emerges as the optimal point, aligning with the findings reported in MDM (Tevet et al., 2023), which further validates the consistency of our results with prior research.

**Failure Case.** We have indeed observed some typical failure cases, as illustrated in Figure 7. In the first example, the model struggles to effectively interpret the prompt related to a "squats twice" motion, resulting in an interruption in the ground truth motion. This is attributed to the rareness of certain words or concepts, which the model may not consistently capture, affecting the alignment

a person jumps sideways to their right several times, then several times to the left.

a person uses their left hand to throw an object in front of them.

the person is walking straight backwards.

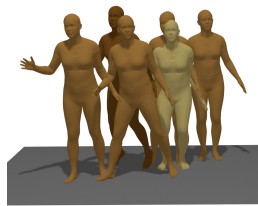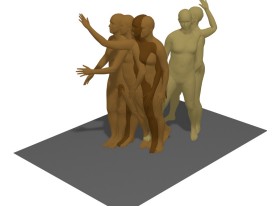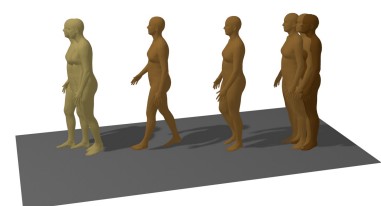

Figure 8: **More visualization of text-to-motion synthesis on HumanML3D dataset**. The gradient of colors, ranging from light to dark, signifies the passage of time.

Table 4: **Evaluation of action-to-motion on the HumanAct12 dataset.** NFE denotes the number of function evaluations. $\rightarrow$ indicates that closer to real is better.

| Method | NFE$\downarrow$ | FID$_{train}$ $\downarrow$ | Accuracy$\uparrow$ | Diversity$\rightarrow$ | Multimodality$\rightarrow$ | #param $\downarrow$ |
|---|---|---|---|---|---|---|
| Real Motion | | $0.050^{\pm.000}$ | $0.990^{\pm.000}$ | $6.880^{\pm.020}$ | $2.590^{\pm.010}$ | |
| Action2Motion (2020) | | $0.338^{\pm.015}$ | $0.917^{\pm.003}$ | $6.879^{\pm.066}$ | $2.511^{\pm.023}$ | |
| ACTOR (2021) | | $0.120^{\pm.000}$ | $0.955^{\pm.008}$ | $6.840^{\pm.030}$ | $2.530^{\pm.020}$ | |
| INR (2022) | | $0.088^{\pm.004}$ | $0.973^{\pm.001}$ | $6.881^{\pm.048}$ | $2.569^{\pm.040}$ | |
| MDM Tevet et al. (2023) | 1,000 | $0.100^{\pm.000}$ | $0.990^{\pm.000}$ | $6.860^{\pm.050}$ | $2.520^{\pm.010}$ | 23M |
| MLD Chen et al. (2023) | 50 | $0.077^{\pm.004}$ | $0.964^{\pm.002}$ | $6.831^{\pm.050}$ | $2.824^{\pm.038}$ | 26.9M |
| **Our MFM** | **10** | $0.110^{\pm.000}$ | $0.978^{\pm.000}$ | $6.850^{\pm.020}$ | $\mathbf{2.610}^{\pm.012}$ | 17.6M |

between the prompt and motion. In the second example, it appears that the model did not effectively interpret the hint from the prompt to turn around and step back, which led to the observed result. We have also observed failure cases in MDM (Tevet et al., 2023) that are comparable or even worse using the same prompt. This underscores that interpreting multiple fine-grained textual descriptions into motion remains a formidable challenge.

**More qualitative results about the text-to-motion synthesis**   We demonstrate more visualization of the text-to-motion synthesis in Figure 8. It shows that the synthesized motion aligns with the prompt textual description.

**More qualitative results about in-between.**   We present our visualization of in-between in Figure 9. Given the prefix and suffix poses as well as the prompt, our model can generate reasonable motions in between.

### C.2 QUANTITATIVE RESULTS

**Result on action-to-motion dataset HumanAct12.**   We present our results on the action-to-motion dataset HumanAct12 in Table 4. Our method achieves results that are comparable to various baselines, all while significantly reducing the number of forward evaluations and utilizing fewer parameters.

**The choice of the text encoder.**   We also explored the possibility of replacing the default CLIP encoder with a more advanced encoder, such as T5 (Raffel et al., 2020) in Table 5. We did not observe an obvious gain in comparison with CLIP embedding. Interestingly, our findings sharply contrast with the conclusions drawn in Imagen (Saharia et al., 2022). This aligns with the results reported in MLD (Chen et al., 2023) (as seen in Table 7 in the Appendix). The disparity in outcomes could potentially be attributed to the current scale of the HumanML dataset, which may not be extensive enough to fully leverage the capabilities of a more sophisticated text encoder like T5.

**Flow matching framework.**   For the sake of completeness, we reiterate the algorithm outlined in Algorithm 2.

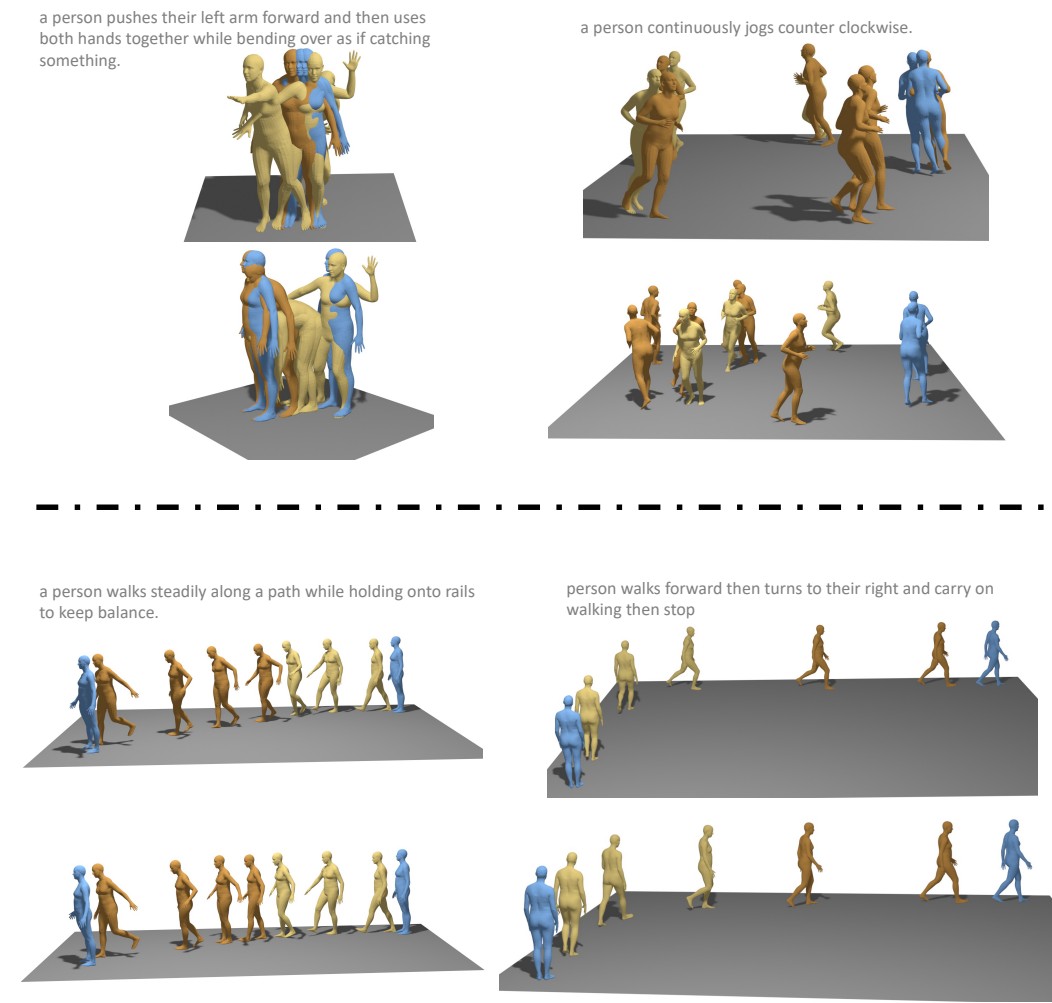

a person pushes their left arm forward and then uses both hands together while bending over as if catching something.

a person continuously jogs counter clockwise.

a person walks steadily along a path while holding onto rails to keep balance.

person walks forward then turns to their right and carry on walking then stop

Figure 9: **More results about trajectory rewriting in in-between editing.** Light blue frames denote motion input, while bronze frames signify generated motion. The gradient of colors, ranging from light to dark, signifies the passage of time.

Table 5: **Ablation study about the text encoder on KIT-ML (Plappert et al., 2016) test set.** $\rightarrow$ indicates that closer to real is better.

| Encoder | FID $\downarrow$ | MM-Dist $\downarrow$ | Diversity $\rightarrow$ | MModality $\uparrow$ |
|---|---|---|---|---|
| Real motion | $0.002^{\pm.000}$ | $2.974^{\pm.008}$ | $9.503^{\pm.065}$ | - |
| CLIP | $0.1822^{\pm.029}$ | $9.04^{\pm.043}$ | $11.06^{\pm.108}$ | $1.53^{\pm.079}$ |
| T5 Raffel et al. (2020) | $0.2301^{\pm.022}$ | $8.97^{\pm.035}$ | $10.951^{\pm.101}$ | $1.52^{\pm.090}$ |

---

**Algorithm 2** Flow Matching Algorithm.

---
1: **Input:** Empirical distribution $q_1$, gaussian distribution $q_0$, batchsize $b$, initial network $v_\theta$.
2: **while** Training **do**
3:    Sample batches of size $b$ *i.i.d.* from the datasets
4:    $\boldsymbol{x}_0 \sim q_0(\boldsymbol{x}_0); \quad \boldsymbol{x}_1 \sim q_1(\boldsymbol{x}_1)$
5:    $t \sim \mathcal{U}(0, 1)$
6:    # Interpolation.
7:    $x_t \leftarrow t\boldsymbol{x}_1 + (1 - t)\boldsymbol{x}_0$
8:    $\text{FM}(\theta) \leftarrow \|v_\theta(x_t, t) - (\boldsymbol{x}_1 - \boldsymbol{x}_0)\|^2$
9:    $\theta \leftarrow \text{Update}(\theta, \nabla_\theta \text{FM}(\theta))$
10: **end while**
11: **Return** FM

---

**Transformer structure.** We illustrate the network structure of our model in Figure 10. We use CLIP (Radford et al., 2021) to encode the text prompt. Given the text embedding **c**, time embedding $t$, and noisy data $x_t$ with position encoding (PE), the transformer is able to ensure the temporal coherence of motion. The resulting output from our model is a vector field, symbolized by $v(\mathbf{x}_t, t, \mathbf{c}; \theta)$, determined at the position $\mathbf{x}_t$ and the timestep $t$. Additionally, our text-to-motion generation is conditioned on CLIP in a classifier-free manner (Ho & Salimans, 2021).

**Python pseudo code.** We provide the pseudo code for trajectory rewriting in Algorithm 3.

---

**Algorithm 3** Euler Sampling algorithm with Trajectory Rewriting.

---
```python
def trajectory_rewriting(model, noise, x1, mask, edit_till = 0.2):
    # x1: origin data.
    # mask: 1 denotes known, 0 denotes unknown.
    # model: pretrained vector field predictor
    z = noise.detach().clone()
    dt = 1.0 / N
    est, traj = [], []

    for i in range(0, N, 1): # fix-step Euler ODE solver
        t = i / N
        x_interp = noise * (1 - t) + x1 * t # interpolate known area.
        if t <= edit_till:
            z = z * (1 - mask) + x_interp * mask # traj rewriting.
        pred = model(z, t) # vector field prediction
        pred = pred.detach().clone()
        _est_now = z + (1 - t)*pred
        est.append(_est_now)

        z = z.detach().clone() + pred * dt
        traj.append(z.detach().clone())

    return traj[-1], est
```

---

# D IMPLEMENTATION DETAIL

## D.1 TRAINING

As the default text encoder, we employ CLIP (Radford et al., 2021), and our flow matching model comprises 8 layers in both the transformer encoder. The feed-forward networks have an output dimensionality of $d_{\text{ff}} = 1024$, while the attention mechanisms employ an inner dimensionality of $d_{\text{kv}} = 512$ with 4 heads. For training, all our models use the AdamW optimizer. The motion tokenizers are trained with a learning rate of $10^{-4}$ and a mini-batch size of 256. Training is conducted on 4 Tesla A5000 GPUs, taking approximately 3 days for HumanML3D and 2 days for KIT-ML.

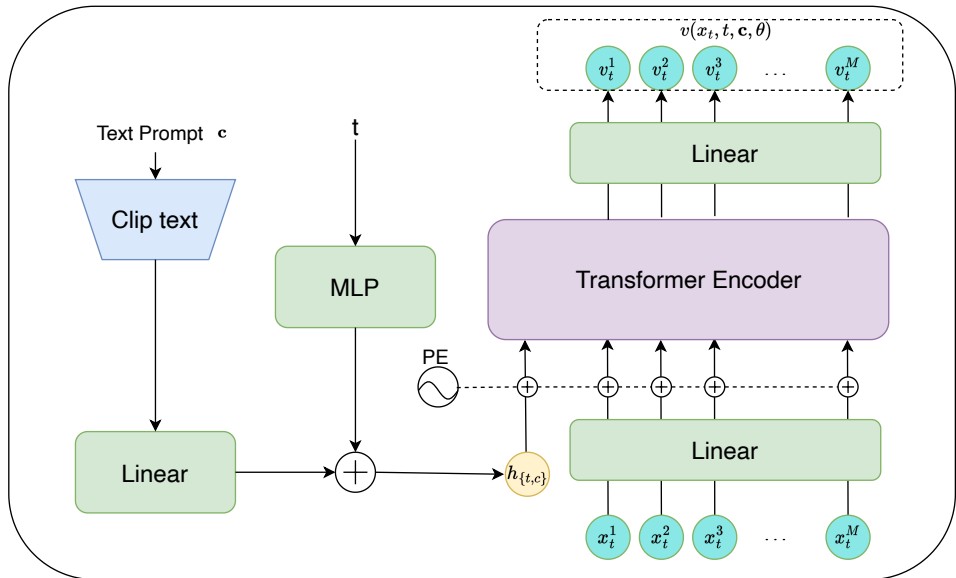

Figure 10: **The dynamic of the flow matching: transformer architecture of text-to-motion generation.** To model the temporal consistency of motion, we employ a transformer architecture. Our input comprises a text prompt $\mathbf{c}$, timestep $t$, and noisy data $\mathbf{x}_t$, which are combined as $\mathbf{h}_{[\mathbf{x}_t,\mathbf{c},t]}$. The output of our model is an estimated vector field denoted as $v(\mathbf{x}_t, t, \mathbf{c}; \theta)$, calculated at the position $\mathbf{x}_t$ and timestep $t$. PE represents the position encoding used in our model. Our text-to-motion generation is conditioned on CLIP in a classifier-free manner (Ho & Salimans, 2021).

In the case of text-to-motion tasks, we encode the textual prompts into $\mathbf{c}$ utilizing a text encoder. The selection of the encoder may be based on preference. Conversely, for action-to-motion tasks, we employ pre-learned embeddings tailored to each distinct class $\mathbf{c}$.

## D.2 EDITING

For trajectory rewriting, we opt for a slightly larger value of $N = 30$, but restrict the editing to the initial $t = 0.2$ timesteps, effectively modifying the first 6 timesteps only.

## D.3 DATASET

**KIT Motion-Language (KIT-ML).** KIT-ML (Plappert et al., 2016) contains 3,911 human motion sequences and 6,278 textual annotations. The total vocabulary size, that is the number of unique words disregarding capitalization and punctuation, is 1,623. Motion sequences are selected from KIT (Mandery et al., 2015) and CMU datasets but downsampled into 12.5 frame-per-second (FPS). Each motion sequence is described by from 1 to 4 sentences. The average length of descriptions is approximately 8. Following Guo et al. (2022a;b), the dataset is split into training, validation, and test sets with proportions of 80%, 5%, and 15%, respectively. We select the model that achieves the best FID on the validation set and reports its performance on the test set.

**HumanML3D.** HumanML3D (Guo et al., 2022a) is currently the largest 3D human motion dataset with textual descriptions. The dataset contains 14,616 human motion and 44,970 text descriptions. The entire textual descriptions are composed of 5,371 distinct words. The motion sequences are originally from AMASS (Mahmood et al., 2019) and HumanAct12 (Guo et al., 2020) but with specific pre-processing: motion is scaled to 20 FPS; those that are longer than 10 seconds are randomly cropped to 10-second ones; they are then re-targeted to a default human skeletal template and properly rotated to face Z+ direction initially. Each motion is paired with at least 3 precise textual descriptions. The average length of descriptions is approximately 12. According to Guo et al. (2022a), the dataset is split into training, validation, and test sets with proportions of 80%, 5%, and 15%, respectively. We

select the model that achieves the best FID on the validation set and reports its performance on the test set.

**HumanAct12.** To further validate the effectiveness of our method, we also test it on the action-to-motion generation task. Action-to-motion involves generating motion sequences based on input action classes, typically represented as scalar values. We employ the well-established benchmark dataset, HumanAct12 (Guo et al., 2020), which serves as the standard evaluation benchmark for action-to-motion models. We evaluate our model using the set of metrics suggested by Guo et al. (2020), namely the Fréchet Inception Distance (FID), action recognition accuracy, diversity, and multimodality. The combination of these metrics constitutes a comprehensive measure of the realism and diversity of the generated motions.

HumanAct12 (Guo et al., 2020) offers approximately 1200 motion clips, organized into 12 action categories, with 47 to 218 samples per label. We adhere to the cross-subject testing protocol used by current works, with 225-345 samples per action class. For both datasets we use the sequences provided by Petrovich et al. (2021).

In the case of action-to-motion, the only change would be the substitution of the text embedding by an action embedding. Since action is represented by a scalar, its embedding is fairly simple; each input action class scalar is converted into a learned embedding of the transformer dimension.

Consistent with Guo et al. (2022a), the maximum motion sequence length on both datasets is 196, while the minimum lengths are 40 for HumanML3D (Guo et al., 2022a) and 24 for KIT-ML (Plappert et al., 2016). For both the HumanML3D (Guo et al., 2022a) and KIT-ML (Plappert et al., 2016) datasets, the motion sequences are truncated to $T = 60$ during training.

# E   MORE DETAILS ON THE EVALUATION METRICS AND THE MOTION REPRESENTATIONS.

## E.1   EVALUATION METRICS

To enhance clarity and comprehensibility for our readers, we reiterate the descriptions of the relevant metrics here to prevent any potential confusion.

We employ standard metrics as defined in Guo et al. (2022a) and Tevet et al. (2023) for our evaluations. To ensure robustness, we repeat each evaluation 20 times and report the average values along with 95% confidence intervals.

We detail the calculation of several evaluation metrics, which are proposed in Guo et al. (2022a). We denote ground-truth motion features, generated motion features, and text features as $f_{gt}$, $f_{pred}$, and $f_{text}$. Note that these features are extracted with pretrained networks in Guo et al. (2022a).

**FID.** FID is widely used to evaluate the overall quality of the generation. We obtain FID by

$$\text{FID} = \|\mu_{gt} - \mu_{pred}\|^2 - \text{Tr}(\Sigma_{gt} + \Sigma_{pred} - 2(\Sigma_{gt}\Sigma_{pred})^{\frac{1}{2}}) \tag{9}$$

where $\mu_{gt}$ and $\mu_{pred}$ are mean of $f_{gt}$ and $f_{pred}$. $\Sigma$ is the covariance matrix and Tr denotes the trace of a matrix.

**MM-Dist.** MM-Dist measures the distance between the text embedding and the generated motion feature. Given N randomly generated samples, the MM-Dist measures the feature-level distance between the motion and the text. Precisely, it computes the average Euclidean distances between each text feature and the generated motion feature from this text:

$$\text{MM-Dist} = \frac{1}{N}\sum_{i=1}^{N}\|f_{pred,i} - f_{text,i}\| \tag{10}$$

where $f_{pred,i}$ and $f_{text,i}$ are the features of the i-th text-motion pair.

**Diversity.**   Diversity measures the variance of the whole motion sequences across the dataset. We randomly sample $S_{dis}$ pairs of motion and each pair of motion features is denoted by $f_{pred,i}$ and $f'_{pred,i}$. The diversity can be calculated by

$$\text{Diversity} = \frac{1}{S_{dis}} \sum_{i=1}^{S_{dis}} ||f_{pred,i} - f'_{pred,i}|| \tag{11}$$

In our experiments, we set $S_{dis}$ to 300 as Guo et al. (2022a).

**MModality.**   MModality measures the diversity of human motion generated from the same text description. Precisely, for the i-th text description, we generate motion 30 times and then sample two subsets containing 10 motions. We denote features of the j-th pair of the i-th text description by $(f_{pred,i,j}, f'_{pred,i,j})$. The MModality is defined as follows:

$$\text{MModality} = \frac{1}{10N} \sum_{i=1}^{N} \sum_{j=1}^{10} ||f_{pred,i,j} - f'_{pred,i,j}|| \tag{12}$$

**Multimodal Distance.**   We calculate the multimodal distance as the average Euclidean distance between the motion feature of each generated motion and the text feature of its corresponding description in the test set. A lower value implies better multimodal distance.

**R Precision. (top-3)**   For each generated motion, R Precision assesses how relevant the generated motions are to the given input prompts. Based on the ground-truth text, a batch of mismatched descriptions is randomly selected from the test set. We calculate the Euclidean distance between the motion feature and text feature of each description in the pool. We count the average accuracy at the top 3 places. If the ground truth entry falls into the top 3 candidates, we treat it as True Positive retrieval. We use a batch size of 32 (i.e. 31 negative examples).

### E.2   MOTION REPRESENTATION

We use the same motion representation as   Guo et al. (2022a).  Each pose is represented by $(\dot{r}^a, \dot{r}^x, \dot{r}^z, r^y, j^p, j^v, j^r, c^f)$, where $\dot{r}^a \in \mathbb{R}$ is the global root angular velocity along the Y-axis; $\dot{r}^x \in \mathbb{R}, \dot{r}^z \in \mathbb{R}$ are the global root velocity in the X-Z plane; $j^p \in \mathbb{R}^{3j}, j^v \in \mathbb{R}^{3j}, j^r \in \mathbb{R}^{6j}$ are the local pose positions, velocity and rotation with $j$ the number of joints; $c^f \in \mathbb{R}^4$ is the foot contact features calculated by the heel and toe joint velocity.

