# OpenReview forum: "Motion Flow Matching for Efficient Human Motion Synthesis and Editing"
_ICLR.cc/2024/Conference — ICLR 2024 Conference Withdrawn Submission_

### Official Review · Reviewer_xr7a · 2023-10-29

**Soundness:** 3 good
**Presentation:** 2 fair
**Contribution:** 2 fair
**Rating:** 5
**Confidence:** 3

**Summary:**

This paper is the first work to combine flow matching with motion synthesis. Moreover, the paper presents "trajectory rewriting", a technique that allows for applications such as motion interpolation.

**Strengths:**

- This is the first work that combines flow matching and motion synthesis, aiming at faster generation, which is a sound motivation.
- The method attains competitive results on KIT.
- A diverse range of tasks have been tested with the proposed framework to show its flexibility.

**Weaknesses:**

- To what extent is the statement "diffusion models are hindered by sampling speed" still true in the current context? The latest works no longer require 1000 steps (e.g., ReMoDiffuse [A] uses only 50 steps, and FLAME [B] has an experiment showing reducing steps does not significantly worsen performance). Some discussions will be helpful.

- I'd appreciate a clarification that the job of ODE is to compute the numerical integral for each step in Line 7 and 9 of Algorithm 1?

- It is highlighted that the proposed framework supports motion editing. However, from my understanding, motion editing typically requires certain conditional (e.g., text) input. In this work, the upper-body editing seems to only mask the upper-body joint features, without any controllability. How would this be considered editing?

- Regarding the experiment, the proposed method archives competitive results on KIT in terms of FID, but not other metrics or on HumanML3D. It will be helpful to discuss why is this the case. It is perfectly fine to not attain SOTA on everything, but studying the limitations can provide the community with key insights.

- Minor: how would late-stage (apply at late time steps) trajectory rewriting compared to the current setting?

- Minor: would "trajectory" rewriting be a misleading name in the context of motion synthesis? This is because "trajectory" (here it means the root translation) is also a critical attribute of synthesized motion.

[A] Zheng et al., ReMoDiffuse: Retrieval-Augmented Motion Diffusion Model

[B] Kim et al., FLAME: Free-form Language-based Motion Synthesis & Editing

**Questions:**

Please see weaknesses.

---

### Official Review · Reviewer_QsSx · 2023-10-30

**Soundness:** 2 fair
**Presentation:** 3 good
**Contribution:** 2 fair
**Rating:** 3
**Confidence:** 4

**Summary:**

This paper aims to address the problem of human motion generation using the more efficient flow matching-based method. The proposed method only requires 10-step sampling to generate a human motion sequence, comparing to hundreds of steps in traditional diffusion models. Moreover, this approach enables a few motion editing applications such as motion in-between/prediction. Quantitative evaluations show that this approach achieves state-of-the-art performance on KIT-ML dataset.

**Strengths:**

1. This work introduces some new insight for human motion synthesis. This is one of few works that use flow match-based approach for motion generation.
2. The propose approach shows considerable flexibiiity and efficiency. Firstly, it enables a number of editing applications, such as motion prediction and interpolation. Secondly, it requires much less steps (10 vs >100) for generation.
3. Experiment shows that this work have SToA performance on one of the human motion benchmarks.
4. The authors have detailed descriptions of the datasets, metrics, implementation, inference/training procedures, as well as their experiment setting.
5. I appreciate the acknowledgement of limitations and failure cases of the proposed approach.

**Weaknesses:**

I am not an expertise in flow matching theory. It took me kind of long time to understand the underlying mathematics. I agree that new approach in an existing task, even if it's not the best, can have some novelty. However, I don't think it's acceptable to have no video demonstration of their results. This is human motion generation, I can not evaluate the performance by looking at a bunch of poses.

Secondly, it is weird that the model can perform well on KIT-ML, but fail on HumanML3D dataset. HumanML3D contains motions of higher quantity and diversity. Does it suggest the model is suitable for smaller dataset?

Thirdly, since this paper claims efficiency as one of their primary goal, it's importance to have a comparison of inference time cost. Even if it requires less setps, does each step actually requires more computations? We don't know. On the other hand, now diffusion model can be efficient as well with DDIM sampling. For example, MLD can generate a motion sequence in 50 steps.

Lastly, it seems this work does not have other novel technical points than flow matching. Given the lack of qualitative results, I feel the significance is undermined.

**Questions:**

Please referring to the weakness section described above.

---

### Official Review · Reviewer_AfYn · 2023-10-31

**Soundness:** 2 fair
**Presentation:** 2 fair
**Contribution:** 2 fair
**Rating:** 5
**Confidence:** 2

**Summary:**

The paper presents a novel human motion generation method based on flow matching, which finds an optimal trade-off between generation quality and the number of sampling steps. The paper proposes a novel training-free motion editing method supporting various downstream applications. Experiments show that the method achieves competitive results over baselines on three datasets.

**Strengths:**

(1) The paper is the first to design a flow-matching-based method for human motion generation, which introduces a novel path to address this task.

(2) The proposed motion editing strategy is a generic method that can support different applications with different types of inputs.

**Weaknesses:**

(1) The paper proposes the first flow-matching-based solution. However, I think the insight of the method should be better explained. Currently, I am confused about whether the method is a direct application of flow matching without task-specific designs.

(2) The evaluation metrics lack examination for method efficiency. I think NFE is not sufficient because the speed of the network forward process is different for each method. The average inference time should be evaluated.

(3) For motion generation, the proposed method does not outperform MLD with the same NFE. The superiority of the method should be better demonstrated.

(4) Minor issues:

* The signal "[]" is unclear in the definition of $p_t$ and Equation (7).

* The signals $x_t$ and $x$ are not defined in Equation (2) and (3).

**Questions:**

In Equation (6), the target vector field is linearly blended by the conditional and the unconditional samplings. I wonder whether the two samplings could contradict to each other during motion generation, for example, one to stand up and the other to sit down due to different goals. If it happens, will the sampling strategy also provide reasonable results?